# Analysis of Host-Specific Differentiation of *Puccinia striiformis* in the South and North-West of the European Part of Russia

**DOI:** 10.3390/plants10112497

**Published:** 2021-11-18

**Authors:** Elena Gultyaeva, Ekaterina Shaydayuk, Philipp Gannibal, Evsey Kosman

**Affiliations:** 1Laboratory of Mycology and Phytopathology, All Russian Institute of Plant Protection, Pushkin, 196608 St. Petersburg, Russia; eshaydayuk@bk.ru (E.S.); fgannibal@vizr.spb.ru (P.G.); 2George S. Wise Faculty of Life Sciences, Institute for Cereal Crops Research, School of Plant Sciences and Food Security, Tel Aviv University, Tel Aviv 69978, Israel; kosman@tauex.tau.ac.il

**Keywords:** *Triticum aestivum*, *Triticum durum*, triticale, *Yr* genes, stripe rust, yellow rust, virulence

## Abstract

Yellow (stripe) rust, caused by *Puccinia striiformis* Westend. (*Pst*), is a major disease of cereals worldwide. We studied *Pst* virulence phenotypes on *Triticum aestivum*, *Triticum durum*, and triticale in three geographically distant regions of the European part of Russia (Dagestan and Krasnodar in North Caucasus, and Northwest) with different climate and environmental conditions. Based on the set of twenty differential lines, a relatively high level of population diversity was determined with 67 different *Pst* pathotypes identified among 141 isolates. Only seven pathotypes were shared by at least two hosts or occurred in the different regions. No significant differentiation was found between regional *Pst* collections of pathotypes either from *T. aestivum* or from *T. durum*. A set of *Pst* pathotypes from triticale was subdivided into two groups. One of them was indistinguishable from most durum and common wheat pathotypes, whereas the second group differed greatly from all other pathotypes. All sampled *Pst* isolates were avirulent on lines with *Yr5*, *Yr10*, *Yr15*, and *Yr24* genes. Significant variation in virulence frequency among all *Pst* collections was observed on lines containing *Yr1*, *Yr3*, *Yr17*, *Yr27*, and *YrSp* genes and cvs Strubes Dickkopf, Carstens V, and Nord Desprez. Relationships between Russian regional collections of *Pst* from wheat did not conform to those for *P. triticina*.

## 1. Introduction

Yellow (stripe) rust, caused by *Puccinia striiformis* Westend. (*Pst*), is a major disease of cereals worldwide. The causative agent of yellow rust infects more than 20 species of cultivated and wild cereals but is more damaging for wheat, triticale, barley, and rye [1,2]. Yield losses from *Pst* infections are usually the result of reduced kernel number per spike, low test weight, and reduced kernel quality [3]. *Pst* is capable of long-distance dispersion by wind movement and human-assisted transport [4]. Urediniospores of *Pst* can be efficiently air-dispersed over hundreds and perhaps thousands of kilometers despite their vulnerability to environmental factors, such as ultraviolet light [1,5]. The spread of urediniospores over very long distances between regions and continents in one step has led to several exotic incursions and yellow rust epidemics at global scale [6].

Yellow rust is considered to be a low-temperature disease and frequently occurs in temperate areas with cool and moist weather conditions. Recent devastating epidemics have occurred in warmer areas where the disease was previously infrequent or absent. This determined that populations of *Pst* had developed adaptation to higher temperatures [1,2,6]. Worldwide analyses of the pathogen population have revealed a clonal structure in northwestern Europe, North and South America, western Australia, and the Mediterranean region. The center of origin for *Pst* was earlier assumed to be Transcaucasia, where grasses were the primary host, and from there the pathogen dispersed in all directions. A recombinant population structure was detected in the regions in and near the Himalayas, identifying this area as a center of diversity [1,6].

In the past two decades, a series of severe epidemics of yellow rust have been reported worldwide, including Europe, North and South America, western Australia, East, Central, and West Asia, and South and North Africa [1,2,4,6]. Since 2000, new races, as well as aggressive and high temperature-adapted strains, have spread the major *Pst* populations worldwide.

In Europe, the established *Pst* population has largely been replaced since 2005. A triticale-aggressive race, first detected in 2006, caused yield losses up to 100% in triticale in Scandinavia, where the epidemics were often difficult to control particularly for organic growers. In 2011, two new races, termed Warrior (*Pst7*) and Kranich (*Pst8*), were detected on both wheat and triticale in many European countries (www.wheatrust.org, accessed on 10 November 2021). The Warrior race is virulent to Avocet lines with genes *Yr1*, *Yr2*, *Yr3*, *Yr4*, *Yr6*, *Yr7*, *Yr9*, *Yr17*, *Yr25*, and *Yr32* and cultivars (cvs) Spalding Prolific (Sp), Avocet S (AvS), and Ambition (Amb). The Kranich race differed from it by avirulence to *Yr4* and cv. Spalding Prolific and virulence to *Yr8*. Isolates of the two races had a number of unusual traits compared to typical isolates from European populations, e.g., by causing more disease on adult plants of wheat genotypes carrying long-term effective adult plant resistance and less disease on others, including previously susceptible genotypes. Hovmøller et al. [6], based on microsatellite genotyping, revealed that Warrior and Kranich races originated from sexually recombining populations in the center of diversity of the yellow rust fungus in the near-Himalayan region of Asia. The Warrior and Kranich races resulted in a continuing replacement of the races that were typical of the European population before 2011 [6]. According to Global Rust Research Center (GRRC) (Aarhus University, Denmark) annual report in 2018–2020, genetic groups *PstS7* (Warrior race) and *PstS8* (Kranich race) were less prevalent than in previous years. Up to now, *PstS7* has been detected in Europe, northern Africa, and South America, and *PstS8* has only been detected in Europe (https://agro.au.dk/forskning/internationale-platforme/wheatrust, accessed on 10 November 2021).

In Russia, yellow rust is a wheat disease of regional significance. The disease is most frequently destructive in the North Caucasus. Epidemics have occurred three or four times over ten years. Under favorable conditions, yield loss can reach 20–30% depending upon the crop growth stage, disease severity, and susceptibility of cultivars [7]. The disease is also regularly observed in the Leningrad region, located in the north-west of Russia [8,9]. In other Russian regions, yellow rust occurs sporadically and was not considered economically important [10,11]. However, global climate change can result in ubiquitous alteration of the disease incidence and severity [12,13].

The cultivation of varieties resistant to yellow rust is one of the most efficient environmentally friendly methods of disease management. However, the rapid development of new virulent races in the pathogen population can overcome resistance in cultivars and often results in high yield loss. Identifying races in *Pst* populations and monitoring distribution and frequency changes of virulences toward *Yr* genes are essential for the effective control of yellow rust [14] and important for selecting resistant cultivars for growing in specific geographic regions and for developing new cultivars with high durable resistance against the pathogen. Analyses of the structure and diversity of *Pst* populations were traditionally based on avirulence/virulence data, and these studies allowed for a rather uniform appreciation of the population characteristics all around the world [15]. The yellow rust pathogen has been characterized with virulence and molecular markers in many world regions [1,6,14,16,17,18,19]. Virulence genes are characterized phenotypically by testing *Pst* isolates on a set of cultivars or single resistance gene lines. In the United States, *Pst* races were characterized using a set of 20 wheat cultivar differentials until 2009. Most cultivars used in the original differential set have two or more genes for stripe rust resistance, which made it difficult to identify virulence to a particular resistance gene. *Yr* near-isogenic lines (NILs) were developed in Australia in the beginning of 2000. Since 2010, a new set of 18 wheat lines carrying single *Yr* genes has been established and used to differentiate races of *Pst*. Using this set of *Yr* single-gene line differentials, more than 300 *Pst* races have been identified from historical and recent collections from the United States and other countries [14,17]. In the GRRC, for the determination of virulence pathotypes, 16 wheat lines carrying single *Yr* genes and three cultivars, Spalding Prolific (Sp), Avocet S (AvS), and Ambition (Amb), are used. Until now, using differential set and SSR genotyping, 17 *Pst* genetic groups were revealed among all world isolate collections (https://agro.au.dk/forskning/internationale-platforme/wheatrust, accessed on 10 November 2021).

In Russia, only the *Pst* population originating from common wheat in the North Caucasus region (Krasnodar, Stavropol, and Rostov) has been studied [20,21,22,23,24], but the presentation of Russian *Pst* isolates in international collections was very limited (www.wheatrust.org, accessed on 10 November 2021). Virulence analysis of the *Pst* population in other regions and from different host species has not been performed yet. The present research was designed to study *Pst* virulence phenotypes in geographically distant Russian regions (North Caucasian and Northwestern) and on different plant hosts (*Triticum aestivum*, *T. durum*, and triticale). The objectives were to (i) assess the virulence variability and race composition of *Pst* isolates infecting common and durum wheat and triticale in distant Russian locations, and (ii) compare the pathogen structure in different host-specific and regional populations.

## 2. Results

### 2.1. Virulence Characterization

In total, 141 *Pst* isolates collected in geographically distant regions of Russia from common and durum wheat and triticale were tested. The number of sampled isolates varied from one (Dagestan collection from triticale) to 41 (Northwestern collection from common wheat). The single isolate sampled from triticale in Dagestan as well as some other groups of isolates with limited sample sizes were included only for the comparison of virulence phenotypes, and were not considered as representative samples of the corresponding populations.

All sampled *Pst* isolates were avirulent on lines with *Yr5*, *Yr10*, *Yr15*, and *Yr24* genes (Table 1). The highest virulence frequencies (≥70%) were observed for resistance genes *Yr6*, *Yr7*, and *Yr8* and cultivars (cvs) Heines VII and Suwon 92/Omar. Significant variation in virulence frequency among all *Pst* collections was observed on lines containing *Yr1*, *Yr3*, *Yr17*, *Yr27*, and *YrSp* genes and cvs Strubes Dickkopf, Carstens V, and Nord Desprez. Virulence frequencies on the *Yr9* gene and cv. Heines Peko were extremely high (>85%) on durum and common wheat, but significantly smaller on triticale (58.3% and 36.1%, respectively). Considerable differences between regional geographically distant *Pst* collections were detected on Avocet lines with *Yr1* and *Yr17* genes, and cvs Strubes Dickkopf and Nord Desprez.

### 2.2. Pathotype Composition

Based on the set of twenty differential lines, 67 different *Pst* virulence phenotypes (pathotypes) were identified among 141 isolates (Table 2, Figure 1): 9, 43, and 20 pathotypes from durum wheat, common wheat, and triticale, respectively; 9, 22, and 47 from the Krasnodar, Dagestan, and North-West regions, respectively. Descriptive parameters of *Pst* phenotypes on wheat and triticale are presented in Table 2. Only four pathotypes were shared by at least two hosts. Predominant pathotype P1 was detected in all three hosts. Two pathotypes (P5 and P6) were found only in North Caucasian collections (Dagestan and Krasnodar) from durum wheat. Pathotype P2 was identified in all three regional collections from common wheat. The average virulence complexity (AVC) of phenotypes originating from triticale (8.7) was significantly smaller than for common and durum wheat (11.1 and 11.7, respectively).

### 2.3. Variability within and among Regional and Host-Specific Collections of Pst Pathotypes

Relationships between all 67 detected *Pst* pathotypes originating from wheat (*T. aestivum* and *T. durum*) and triticale are shown in the UPGMA dendrogram (Figure 1). Most pathotypes (60) could not be distinguished either by the region or by the host of origin, even though a separation into two big unidentified clusters of mixed composition seems obvious. Nevertheless, there is clear divergence of a group of nine pathotypes in the bottom branch of the dendrogram from all other pathotypes (Figure 1). This group of ‘outliers’ consisted of two very different sets of seven Northwestern pathotypes from triticale and two Dagestan pathotypes from common wheat.

Since some *Pst* pathotypes from triticale were similar to those from *T. aestivum* and *T. durum* (main cluster in Figure 2), while others belonged to the group of ‘outliers’, we separately analyzed relationships between the triticale pathotypes. Two very distinct sets of *Pst* pathotypes from triticale were revealed (see the UPGMA dendrogram in Figure 2). No clear split into clusters was obtained for the host-specific pathotypes either from common or from durum wheat (data not shown). The regional groups of *Pst* pathotypes from Dagestan and Krasnodar were not subdivided either, whereas the same triticale ‘outliers’ formed a separate cluster in the pathotype collection from the North-West region (data not shown).

Based on the findings regarding the relationships between *Pst* pathotypes, four host-specific collections of pathotypes were further considered: *T. aestivum* (Ta), *T. durum* (Td), and two groups of triticale pathotypes (Tr1 and Tr2). The Tr2 collection of pathotypes from triticale was the most distant (*KB* values range from 0.224 to 0.262) and significantly different (p<0.01) from all others (Table 3), whereas a hypothesis of no differentiation between all pairs of Ta, Td, and Tr1 collections was not rejected (i.e., these three collections were indistinguishable). The highest variability was within the collection from *T. aestivum*, whereas the lowest one was within the Tr1 collection from triticale (*KW* = 0.291 and 0.15, respectively).

A notable difference between the two *Pst* collections from triticale was clearly illuminated by infection type reactions on several *Yr* genes (Table 4). Thus, virulence frequency to *Yr3*, *Yr4*, *Yr9*, *YrSD*, *Yr27*, and *Yr6* genes was low and much smaller in the Tr2 collection, whereas all pathotypes of the Tr1 collection were virulent on *Yr3*, *Yr4*, *Yr9*, and *Yr27*.

The most diverse regional collection of *Pst* pathotypes was the North-West one (*KW* = 0.357), perhaps due to the fact that the largest number of isolates (80) was tested in this region, and it included all but one of the triticale pathotypes (19) along with those from *T. aestivum* (25) and *T. durum* (3). The North-West regional *Pst* collection did not differentiate significantly from the two geographically distant collections from Krasnodar and Dagestan.

Finally, we compared six host-specific collections from different regions: three regional collections from *T. aestivum* (Kr_Ta, D_Ta, and NW_Ta), the *T. durum* collection from Dagestan (D_Td), and two triticale collections from the North-West region (NW_Tr1 and NW_Tr1); each collection consisted of five pathotypes at least. Relationships between these collections based on the *KB* distance with regard to the simple mismatch dissimilarity between them are expressed in the UPGMA dendrogram in Figure 3. The second group of triticale pathotypes NW_Tr2 was far apart from all other rather similar groups (*KB* distance between NW_Tr2 and all other groups varied from 0.219 to 0.283, whereas the pairwise distances between other five groups were in the range of 0.114–0.171).

## 3. Discussion

The studied isolates of *Pst* were collected from common wheat, durum wheat, and triticale in three geographically distant regions of the European part of Russia with considerably different climates. The distance between Saint Petersburg (North-West) and Derbent (Dagestan) is about 2600 km, between Saint Petersburg and Krasnodar (south, western territories of North Caucasus)—1800 km, and between Krasnodar and Derbent (south, eastern territories of North Caucasus)—800 km. The Derbent region of Dagestan belongs to the coastal zone, and the climate is transitional from temperate maritime to subtropical semi-dry. The climate of the Krasnodar territory is moderate continental. There are almost always hot summers, mild snowless winters, and little rainfall. The Leningrad region (North-West) belongs to a zone of temperate climate, transitioning from oceanic to continental, with moderately mild winters and moderately warm summers. Winter wheat and triticale cultivars are grown in the Dagestan and Krasnodar regions, whereas winter and spring wheats and triticale are cultivated in the North-West region.

Virulence analysis of *Pst* from common wheat, durum wheat, and triticale indicated a relatively high level of variation within the Russian population of the pathogen, with 67 pathotypes identified among 141 isolates. However, despite the geographical remoteness and differences in climate and growing crops (winter vs. spring wheats), no significant differentiation was found between regional *Pst* collections either from *T. aestivum* or from *T. durum*. These results are consistent with earlier reported weak differentiation among *Pst* populations in geographically distant regions [18,25]. Virulence characterization of international collections of the wheat yellow rust pathogen from 13 countries (Algeria, Australia, Canada, Chile, China, Hungary, Kenya, Nepal, Pakistan, Russia, Spain, Turkey, and Uzbekistan) revealed common and unique virulences and pathotypes in the corresponding *Pst* populations. Virulence phenotypes were generally different among the countries; however, most of the virulences were common among isolates from different countries [25]. Comparative analyses of *Pst* populations from Saskatchewan and southern Alberta (Canada) with the populations from the Great Plains and the Pacific Northwest of the United States indicated the close similarity of races in all the regions [16]. It is important to note that contrary to the *Pst* population, the regional structure of another rust pathogen *P. triticina* in Russia was characterized by substantial differentiation among regions on both *T. aestivum* [26] and *T. durum* [27]. This contrast between the pathogens raises a need to develop different approaches to the monitoring and control of wheat yellow rust and leaf rust epidemics.

We compared Russian *Pst* pathotypes with those from the GRRC survey in 2018–2020 (https://agro.au.dk/forskning/internationale-platforme/wheatrust, accessed on 10 November 2021), where lines with genes *Yr1*, *Yr2*, *Yr3*, *Yr4*, *Yr5*, *Yr6*, *Yr7*, *Yr8*, *Yr9*, *Yr10*, *Yr15*, *Yr17*, *Yr24*, *Yr25*, *Yr27*, and *Yr32* and cvs Spalding Prolific (Sp), Avocet S (AvS), and Ambition (Amb) were used for virulence analysis. Nearly all of these lines and cultivars (except the line with *Yr25* and cv. Ambition) were used in our study. We revealed races similar to those reported in the GRRC survey. *PstS1*,*v1*,*v27*, *PstS2*,*v3*, *PstS2*,*v27*, *PstS2*,*v1*,*v27*, and *PstS3* phenotypes were determined among the northwestern isolates from triticale. Note, however, that using two SCAR markers for tracking the distribution and origin of invasive strains [28], we did not determine *PstS1* and *PstS2* strains among the northwestern isolates from triticale in our previous study [29].

We found that pathotypes sampled from common and durum wheat and some pathotypes from triticale were similar to a large extent. Such low pathogen distinction between virulence phenotypes identified on different crops was also revealed within *Pst* populations in Ethiopia [30], Lebanon, and Syria [18], and some other countries [2,31]. In most cases, no *Pst* differentiation between host types (common wheat, durum wheat, triticale, and volunteers) was found, though sometimes a considerable difference of triticale isolates from those of wheat origin was noted. In the study of international *Pst* collections from wheat, triticale, and barley [31], the lineage of triticale and barley isolates was separated from all wheat isolates.

Our set of *Pst* pathotypes from triticale was subdivided into two groups. One of them was indistinguishable from most durum and common wheat pathotypes, whereas the second group differed greatly from all other pathotypes. The separation and differentiation of virulence phenotypes (and SSR genotypes) from triticale were also observed in a study of leaf rust (*P. triticina*) in Poland [32]. Note that none of the *Pst* pathotypes identified on triticale were similar to the unique “triticale-aggressive” race (PstS4), which was first discovered in 2006 and has spread and caused severe damage on triticale throughout Europe [6]. This race was virulent to *Yr10* and *Yr24* in contrast to all Russian isolates avirulent on these genes. Yet, two Russian pathotypes from triticale were closely related to the *PstS13* race, determined in Europe in 2015 with only one difference in virulence to either *Yr17* or *Yr32*.

Several breeding methods exist for developing new triticale cultivars. One of them is based on crossing wheat with rye, followed by doubling the number of chromosomes in F1 hybrids using colchicine (primary triticale). Another approach is the development of secondary triticale, which is the result of crossing amphidiploids of different ploidy levels [33]. When individual chromosomes of rye are replaced by chromosomes of the wheat genome D, the phenotypes of the obtained forms become more similar to wheat. This perhaps can explain the similarity of *Pst* pathotypes from triticale and wheat in our and other studies [18,30]. Unfortunately, the lack of information about breeding sources of secondary triticale cultivars does not allow us to check this hypothesis.

Population studies of plant pathogens often refer to two types of diversity: gene/virulence and genotype/pathotype diversity. Gene diversity refers to the numbers and frequencies of alleles at individual loci in a population, whereas genotype diversity refers to the number and frequency of multilocus genotypes. In the case of *P. striiformis* on agricultural crops, virulence and pathotype diversity were considered the most important traits of pathogen variation [31]. The leading population parameters used in our study were the assignment-based metrics of genetic variation within and among populations (a kind of combination of gene and genotype diversity) [34,35,36].

This research is the first large-scale study of *Pst* in Russia. The monitoring of the *Pst* population from common wheat in the south of the European part of Russia (Krasnodar, Rostov, and Stavropol locations) was recently performed by Volkova et al. [23]. They identified avirulence of all collected isolates only to *Yr3*, *Yr5*, *Yr26*, and *YrSP* resistance genes among the tested *Yr* single-gene lines. Our results were a bit different, so that resistance genes *Yr5*, *Yr10*, *Yr15*, and *Yr24* were found to be highly efficient, as in most regions all over the world. Isolates virulent to them were either absent or occurred with a low frequency [18,25,28,37,38].

Genes *Yr2*, *Yr6*, *Yr7*, and *Yr8* were found ineffective in the north-west and south of the European part of Russia. These genes have also lost their effectiveness in some other regions worldwide, including Central Asia, West Asia, North Africa, and North America 16–18,25]. Resistance to the *Yr2* gene was quashed, perhaps due to the fact that this gene is very common in both winter and spring wheat cultivars and common in wheat distributed through CIMMYT [39]. The high frequency of *Yr6* virulence worldwide can be explained by the presence of *Yr6* in common wheat and durum wheat cultivars [40]. Similarly, the high frequency of *Yr7* virulence could be due to the worldwide use of *Yr7* through Thatcher (*Yr7*) and its presence in old cultivars and landraces [39]. The wide distribution and high frequency of *Yr8* virulence could be due to the wide use of this gene from *Aegilops comosa* and its common presence in grasses. Virulence to *Yr9* of all collected isolates from common and durum wheat and most isolates from triticale could be a result of the extensive use of cultivars with *Yr9* originating from *Secale cereale* [41,42]. This gene is linked with leaf *Lr26* and stem *Sr31* rust resistance genes and is widespread in varieties of Russian and world selection.

We established significant differences between the three geographic regions of Russia and between the hosts in virulence to *Yr1*, *Yr17*, *Yr27*, and *YrSp*. Variation in the frequencies of virulence to these genes was also noted in several world regions [16,17,19,25,37]. Virulence to *Yr17* was low or absent in the Russian *Pst* collections except the isolates from durum wheat in Krasnodar. In most studies, a very high virulence frequency to *Yr17* of at least 80% was reported [16,17,25]. The *Yr17* gene is widespread in wheat cultivars grown in Western Europe and North America [14], whereas in Russia, the cultivation of winter wheat with the *Yr17* gene began in the Krasnodar region in 2015, and the corresponding cultivars have been characterized by adult plant resistance to yellow rust [43].

The presence of identical pathotypes in geographically distant regions could be evidence of the long-distance dispersal capacity of the yellow rust pathogen in Russia. The airborne spores contribute to variation in *P. striiformis* far apart from the sites of their origin. Grasses serve as additional reserves of infection. It was shown that yellow rust isolates from grasses successfully infect winter wheat commercial cultivars. Cheng et al. [2] indicated that grasses harbor more diverse *Pst* isolates than cereals. Thus, grasses can also donate to the pathogen variability in both regional and host-specific populations of *P. striiformis*.

## 4. Materials and Methods

### 4.1. Yellow Rust Sampling and Spore Multiplication

Leaves bearing uredinia of yellow rust were collected in 2020–2021 from common and durum wheat in two parts of North Caucasus including the field plots at Dagestan Experiment Station of N.I. Vavilov All-Russian Institute of Plant Genetic Resources (Derbent) and in a commercial field of the Krasnodar region, and in an experimental field of the National Center of Grain named after P.P. Lukyanenko. In the North-West region, infection material was collected during 2019–2021 from common and durum wheat and triticale on the experimental plots of the All Russian Institute of Plant Protection and All-Russian Institute of Plant Genetic Resources (Saint Petersburg, Pushkin), as well as on the State variety testing plots (Leningrad region, Gatchina district) and on commercial fields (Leningrad region, Lomonosovsky district) (Figure 4). Sampling details of *Pst* isolates are shown in Table 5. The rust samples were collected from infected wheat and triticale cultivars grown in the experimental plots or commercial fields. From three to ten leaves of a single variety from each plot/field were considered as one sample. Infected leaves were air-dried and stored at 4 to 5 °C until spores were collected for inoculation and increase. Up to two single uredinial isolates were derived from each rust sample and tested for infection type. In the case of a bulk of infected leaves from commercial fields, virulence phenotypes of at least three single pustule isolates were determined.

Standard procedures and conditions with some modification were used for inoculating and growing plants before and post inoculation, and collecting urediniospores [18,44]. Wheat seeds (10–15 grains) were planted in fill pots with commercial potting soil mix Terra vita (https://terravita.company.site, accessed on 13 November 2021) and grown in a rust-free greenhouse. Seedlings of highly susceptible cv. ‘Michigan Amber’ at the two-leaf stage were inoculated with urediniospores from each *Pst* sample. Prior to inoculation, the leaf samples with *Pst* uredia were incubated at 3–5 °C for 1–2 days in a Petri dish or covered plastic trays to induce fresh urediniospore production. The lower part of the leaf segments was covered with cotton wool with benzimidazole solution (0.004%) (Appendix A). Pieces of single lesions were tied with cling film onto 10–12-day-old wheat seedlings of susceptible cultivars (Appendix A). Inoculated seedlings were incubated in a dew chamber at 10 °C for 24 h in darkness (Appendix A). Then, the film with infectious material was removed and the pots were transferred to a Versatile Environmental Test Chamber MLR-352H (SANYO ElectricCo., Ltd., Osaka, Japan) programmed at a light cycle of 10 °C (8 h in darkness) and 16 °C (16 h of light). Each pot was isolated with perforated cellophane bags to prevent contamination with airborne pathogens (Appendix A). About 14 to 17 days after inoculation, sporulation began on inoculated leaves (Appendix A). Urediniospores were harvested with a cyclone spore collector and used for multiplication (Appendix A).

### 4.2. Virulence Analysis

*Pst* isolates were tested for infection type on a differential set of 12 wheat lines in the Avocet spring wheat background and on 8 supplemental wheat differentials (Table 1). Two cultivars, Jupateco S and Avocet S, were used as susceptible controls. Seeds of all differential lines and cultivars were kindly provided by Prof. A.S. Rsaliev (Scientific Research Institute of Biological Safety Problems, Kazakhstan).

Urediniospores of a single isolate (10–20 mg) were suspended in 5 mL Novec™ 7100 in a glass tube and connected to the airbrush spray gun (Appendix Ah). This suspension was sprayed onto 10–12 two-leaf stage seedlings of each differential line. The inoculated plants were incubated in a dew chamber at 10 °C for 24 hr in the dark and transferred to a controlled-climate chamber with the same parameters that were used for isolate multiplication. Seedling infection type (IT) was determined in 17 days (Appendix Ai), on a scale from 0 to 4 [43], where scores of 0–2 indicated resistant reaction of the host plant (pathogen avirulence) and scores 3–4 indicated plant susceptibility or virulence reaction of a given isolate [39,45].

Repeated tests were conducted for isolates showing virulence patterns different from previously identified races. Two or more repeats were performed with sub- or single-uredium isolates to confirm the virulence or avirulence patterns of races.

### 4.3. Data Analysis

Original virulence data were clone-corrected so that one unique *Pst* virulence phenotype per each selected set of isolates was considered. Such clone-corrected data were further analyzed for structural relationships and variability. Virulence frequency and relative virulence complexity [46] were calculated for all regional sets of phenotypes (Dagestan, Krasnodar, and North-West) and for each group of phenotypes originating from a specific host (*T. aestivum*, *T. durum*, and Triticale).

The relationships among the virulence phenotypes of *Pst* were established with the UPGMA dendrograms on the basis of the simple mismatch dissimilarity. Variability of the *Pst* collections was analyzed with Kosman’s assignment-based metrics of dispersion (*KW*) within, and distance (*KB*) between collections with regard to the simple mismatch dissimilarity [34,35,36]. Significance of differentiation among the *Pst* collections was estimated with the permutation test (500 random partitions) for differentiation statistics based on the *KW* dispersion
$$dif_{KW}=\frac{KW\left(P\right)-\sum_{i=1}^{k}\mu_{i}KW\left(P_{i}\right)}{1-\sum_{i=1}^{k}\mu_{i}KW\left(P_{i}\right)}$$
(Equation (1) in [27]) as was explained in Kosman et al. [34] (p. 565) for a pool *P* of all phenotypes from a set of *k* collections $P_{i}$ of size $n_{i}$, $N=n_{1}+n_{2}+\ldots +n_{k}$, $\mu_{i}=n_{i}/N$ (i=1,2,…,k). The corresponding calculations were performed with VIRULENCE ANALYSIS TOOL (VAT) software [47,48] and its extension.

UPGMA dendrograms with regard to the simple mismatch dissimilarity between virulence phenotypes and the Kosman distance (*KB*) between the host-specific collections of *Pst* phenotypes were derived using the SAHN program of the NTSYSpc v. 2.21s (Exeter software).

## 5. Conclusions

This study was the first attempt of large-scale analysis of *Pst* in Russia. The total virulence variation of the pathogen isolates collected in three geographically distant regions on three different host species was relatively high. *Pst* pathotypes in collections sampled from common and durum wheat and some pathotypes from triticale were similar to a large extent, whereas several other triticale pathotypes were significantly different. Identical pathotypes were determined in geographically distant Russian regions, which could be evidence of the long-distance dispersal capacity of the yellow rust pathogen. The virulence analysis revealed highly efficient and partially effective *Yr* genes.

## Figures and Tables

**Figure 1 plants-10-02497-f001:**
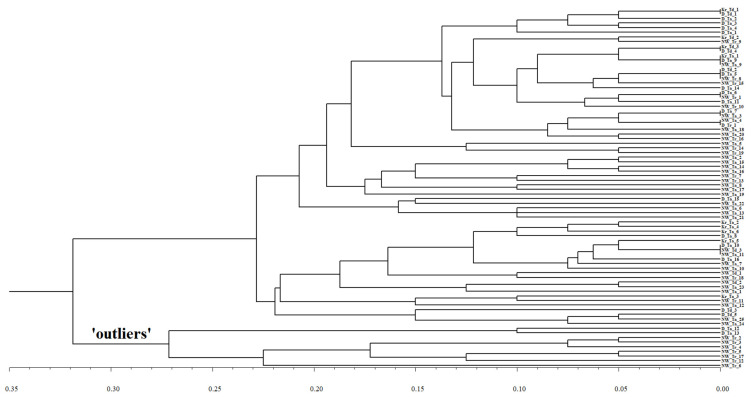
UPGMA dendrogram of relationships among *P. striiformis* pathotypes based on the simple mismatch dissimilarity between them. Abbreviations of regions and host species: D—Dagestan; Kr—Krasnodar; NW—North-West; Ta—T. aestivum; Td—*T. durum*; Tr—triticale. For example, pathotype designation Kr_Td_1 means that this pathotype is number 1 from T. durum in the Krasnodar region. Group of ‘outliers’ is indicated.

**Figure 2 plants-10-02497-f002:**
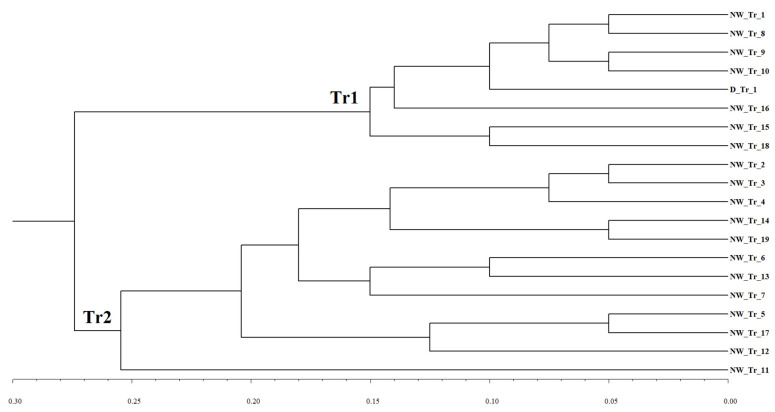
UPGMA dendrogram of relationships among *P. striiformis* pathotypes from triticale based on the simple mismatch dissimilarity between them. Abbreviations of regions and host species: D—Dagestan; NW—North-West; Tr—triticale. For example, pathotype designation NW_Tr_1 means that this pathotype is number 1 from triticale in the North-West region. Two clusters of the triticale pathotypes Tr1 and Tr2 are shown.

**Figure 3 plants-10-02497-f003:**
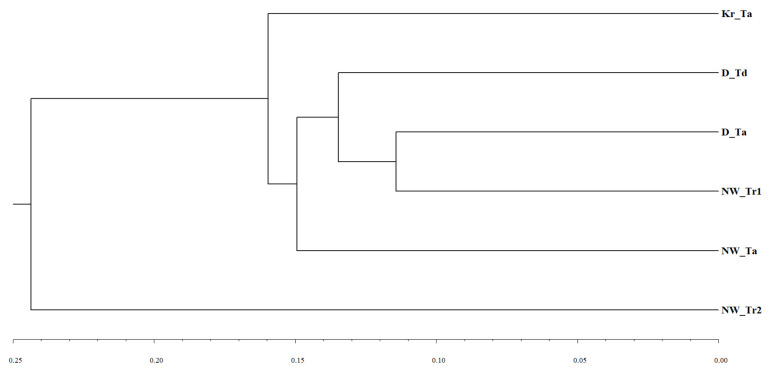
UPGMA dendrogram of relationships among *P. striiformis* collections of pathotypes from different hosts and regions based on the *KB* distance with regard to the simple mismatch dissimilarity between collections. Abbreviations of regions and host species: D—Dagestan, Kr—Krasnodar, NW—Noth-West; Ta—*T. aestivum*, Td—*T. durum*, Tr1 and Tr2—two groups of triticale (see Figure 2). For example, designation Kr_Ta means that this collection is from *T. aestivum* in the Krasnodar region; designation NW_Tr2 means that this collection is from the second group of triticale Tr2 in the North-West region.

**Figure 4 plants-10-02497-f004:**
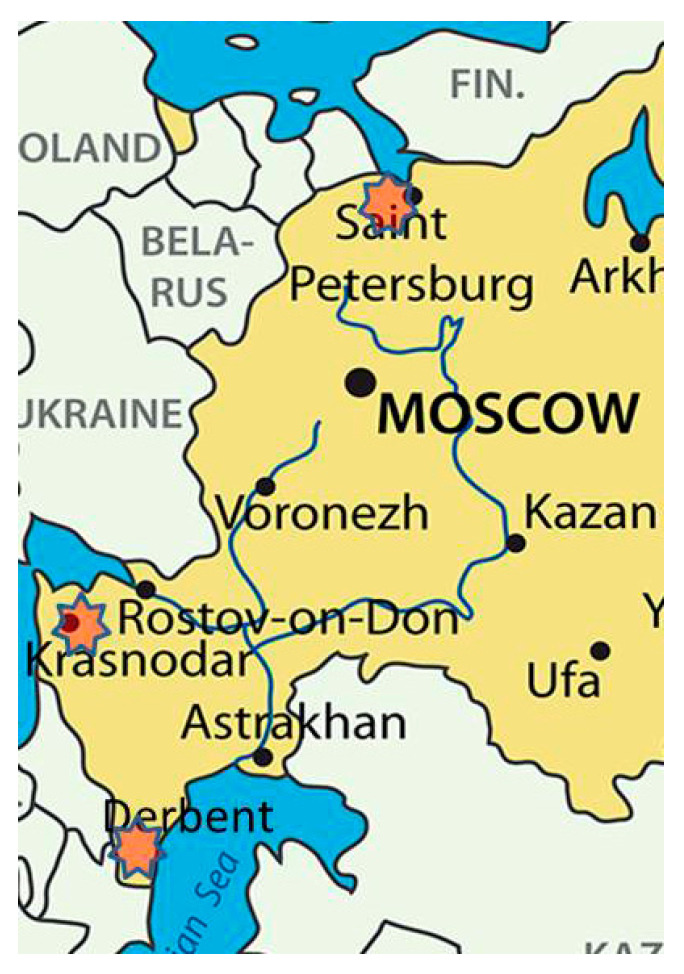
Collection sites of *Puccinia striiformis* in Russia.

**Table 1 plants-10-02497-t001:** Virulence frequency of *P. striiformis* collections of isolates originating from wheat and triticale in Russian regions.

*Yr* Genes	Line Containing the Corresponding *Yr* Genes	*T. durum*				*T. aestivum*				Triticale
		North Caucasus		NW	Total	North Caucasus		NW	Total	NW
		Kr ^a^	D			Kr	D			
*Yr1*	*Yr1*/6*Avocet S, Chinese 166	33.3	70	100	**63.2**	100	46.4	58.5	**62.4**	**11.1**
*Yr2*, *Yr*+ ^b^	Heines VII	100	100	100	**100**	93.8	96.4	100	**97.6**	**91.7**
*Yr3*	Vilmorin 23	66.7	80	100	**78.9**	81.3	67.9	82.9	**77.6**	**47.2**
*Yr4*, *Yr*+	Hybrid 46	100	100	100	**100**	100	92.9	63.4	**80**	**36.1**
*Yr5*	*Yr5*/6*Avocet S,	0	0	0	**0**	0	0	0	**0**	**0**
*Yr6*	*Yr6*/6*Avocet S	100	100	100	**100**	100	96.4	100	**98.8**	**100**
*Yr7*	*Yr7*/6*Avocet S,	100	100	100	**100**	100	100	100	**100**	**100**
*Yr8*	*Yr8*/6*Avocet S	100	70	100	**84.2**	100	100	100	**100**	**100**
*Yr9*	*Yr9*/6*Avocet S	100	100	100	**100**	100	85.7	100	**95.3**	**58.3**
*Yr10*	*Yr10*/6*Avocet S	0	0	0	**0**	0	0	0	**0**	**0**
*Yr15*	*Yr15*/6*Avocet S	0	0	0	**0**	0	0	0	**0**	**0**
*Yr17*	*Yr17*/6*Avocet S	100	30	0	**15.8**	0	0	17.1	**8.2**	**0**
*Yr24*	*Yr24*/6*Avocet S	0	0	0	**0**	0	0	0	**0**	**0**
*YrSD*, *Yr*+	Strubes Dickkopf (SD)	0	10	66.7	**15.8**	12.5	53.6	78	**57.6**	**2.8**
*Yr27*	*Yr27*/6*Avocet S	66.7	100	66.7	**84.2**	56.3	100	70.7	**77.6**	**55.6**
*Yr32*, *Yr*+	Carstens V	66.7	70	33.3	**63.2**	56.3	67.9	75.6	**69.4**	**41.7**
*YrSu*, *Yr+*	Suwon 92/Omar (Su)	100	100	100	**100**	100	92.9	97.6	**96.5**	**100**
*YrSp*	*YrSP*/6*Avocet S	33.3	0	100	**26.3**	37.5	46.4	12.2	**28.2**	**30.6**
*Yr3*, *YrND*, *Yr+*	Nord Desprez (ND)	0	0	33.3	**5.3**	0	10.7	19.5	**12.9**	**13.9**
*Yr6*, *Yr+*	Heines Peko (HP)	100	100	100	**100**	100	96.4	85.4	**91.8**	**36.1**

^a^ Kr—Krasnodar, D—Dagestan (Derbent), NW—North-West (St. Petersburg); ^b^
*Yr+* corresponds to additional unidentified genes.

**Table 2 plants-10-02497-t002:** Pathotypes of *P. striiformis* identified among isolates originating from wheat and triticale in three Russian regions.

Parameters	*T. durum*				*T. aestivum*				Triticale		
Location	Kr ^a^	D	NW	Total	Kr	D	NW	Total	NW	D	Total
Number of isolates	6	10	3	**19**	16	28	41	**85**	36	1	**37**
Number of pathotypes	3	5	3	**9**	6	16	25	**43**	19	1	**20**
Predominant abundances ^b^, %	33.3	40	33.3	**18.2**	43.8	26.8	12.2	**6.4**	11.1	100	**10.8**
Average virulence complexity (AVC)	10.7	11.2	13	**11.7**	11.7	10.7	11.3	**11.1**	8.5	12	**8.7**
Prevailing pathotypes (designation: virulence formula of *Yr* genes), and their abundances (%):											
P1: 2,3,4,6,7,8,9,27,Su,HP ^c^	0	10	0	**5.3**	0	3.6	0	**1.2**	5.6	0	**5.4**
P2: 1,2,3,4,6,7,8,9,27,Su,HP	0	0	0	**0**	43.8	3.6	2.4	**12.4**	0	0	**0**
P3: 1,2,3,4,6,7,8,9,27,Su,HP	0	0	33.3	**1.2**	0	28.6	2.4	**10.6**	0	0	**0**
P4: 2,3,4,6,7,8,9,27,32,Su,HP	0	0	0	**0**	0	7.1	0	**5.4**	5.6	0	**5.4**
P5: 1,2,3,4,6,7,8,9,27,32,Su,HP	33.3	40	0	**26.3**	0	0	0	**0**	0	0	**0**
P6: 2,4,6,7,8,9,27,32,Su,HP	33.3	20	0	**15.7**	0	0	0	**0**	0	0	**0**
P7: 2,3,4,6,7,8,9,27,32,Su,HP	0	0	0	**0**	0	0	7.3	**4.7**	0	100	**2.7**

^a^ North Caucasus: Kr—Krasnodar, D—Dagestan (Derbent), NW—North-West (St. Petersburg). ^b^ Frequency of predominant pathotypes (%). ^c^ Pathotype P1 is virulent on lines with *Yr2*, *Yr3*, *Yr4*, *Yr6*, *Yr7*, *Yr8*, *Yr9*, *Yr27*, *YrSu*, and *YrHP*, and avirulent on the rest of the *Yr* genes tested.

**Table 3 plants-10-02497-t003:** Variation within and among host-specific collections of *P. striiformis* pathotypes in Russia.

Hosts	*T. durum*	*T. aestivum*	Tr1 ^a^	Tr2 ^a^
*T. durum*	0.227 ^b^	**0.114** ^c^	**0.111**	**0.262**
*T. aestivum*	a ^d^	0.291	**0.123**	**0.224**
Tr1	a	a	0.15	**0.233**
Tr2	b ^e^	b	b	0.233

^a^ Tr1 and Tr2—groups of triticale pathotypes established in the UPGMA dendrogram in Figure 2. ^b^
*KW* dispersion within a group of host-specific pathotypes is on diagonal. ^c^
*KB* distance between groups of host-specific pathotypes is above diagonal (bold font). ^d^ *a*—no differentiation among groups of host-specific pathotypes at *p* = 0.1 (likelihood of differentiation is less than 0.9). ^e^ *b*—significant differentiation among groups of host-specific pathotypes at *p* = 0.01 (likelihood of differentiation exceeds 0.99).

**Table 4 plants-10-02497-t004:** Virulence frequencies in two groups of *P. striiformis* collections from triticale in Russia.

	*Yr1*	*Yr2*	*Yr3*	*Yr4*	*Yr9*	*Yr27*	*Yr32*	*YrSp*	SD ^a^	ND ^a^	HP ^a^
Tr1 ^b^	12.5	100	100	100	100	100	25	100	25	12.5	75
Tr2 ^b^	16.7	83.3	25	8.3	50	25	0	25	0	25	16.7

^a^ SD—Strubes Dickkopf, ND—Nord Desprez, HP—Heines Peko. ^b^ Tr1 and Tr2—groups of triticale pathotypes established in the UPGMA dendrogram in Figure 2. Virulence reaction to all other *Yr* genes was monomorphic. All isolates were avirulent to Avocet lines with genes *Yr5*, *Yr10*, *Yr15*, *Yr17*, and *Yr24*, and virulent to *Yr6*, *Yr7*, *Yr8*, and *YrSu* (Suwon 92/Omar).

**Table 5 plants-10-02497-t005:** Sampling details of isolates in regional collections of *P. striiformis*.

Regions (Location)	Year	*T. aestivum*		*T. durum*		Triticale	
		No. of Samples	No. of Isolates	No. of Samples	No. of Isolates	No. of Samples	No. of Isolates
North Caucasus (Krasnodar (Kr))	2020	5	9	4	6	0	0
	2021	5	7	0	0	0	0
	**Total**	**10**	**16**	**4**	**6**	**0**	**0**
North Caucasus (Dagestan, Derbent (D))	2020	4	6	3	3	0	0
	2021	19	22	7	7	1	1
	**Total**	**23**	**28**	**10**	**10**	**1**	**1**
North-West (Saint Petersburg (NW))	2019	2	3	5	8	4	7
	2020	15	24	0	0	17	25
	2021	9	9	0	0	4	4
	**Total**	**26**	**36**	**5**	**8**	**25**	**36**

## Data Availability

Original data are available from the corresponding author upon reasonable request.

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
