# Peer review of "Analysis of Host-Specific Differentiation of *Puccinia striiformis* in the South and North-West of the European Part of Russia"

_plants, 2021, doi:10.3390/plants10112497_

Round 1
Reviewer 1 Report
Chapter 1:
The introduction did not take into account the knowledge about other virulence genes (e.g. Warrior) that are important in Europe and worldwide.
chapter 3: discsussion:
The introduction did not take into account the knowledge about other virulence genes (e.g. Warrior) that are important in Europe and worldwide.
In Chapter 3: Discussion, no comparison was made with other publications on yellow rust studies with pathotypes/breeds (e.g. Warrior) and virulence frequency (e.g. in Central Europe) or the significance of the findings for cultivation in Russia.
Here, the author should update the discussion.
chapter 4:
Can the author explain which method was used to determine the virulence genes of the isolates and the composition of the pathotypes?
Were only ad planta tests used or also molecular biological methods?
For this question, the author is asked to explain and explain the missing information.
Author Response
Dear reviewer, thank you very much. We added some information according to your comments.
With kind regards,
Elena Gultyaeva
Answers to comments.
Chapter 1: The introduction did not take into account the knowledge about other virulence genes (e.g. Warrior) that are important in Europe and worldwide.
Introduction was considerably extended and relevant information was added (lines 44-82, 105-124)
Chapter 3: Discussion, no comparison was made with other publications on yellow rust studies with pathotypes/breeds (e.g. Warrior) and virulence frequency (e.g. in Central Europe) or the significance of the findings for cultivation in Russia. Here, the author should update the discussion.
Discussion was updated (lines 292-304)
Chapter 4: Can the author explain which method was used to determine the virulence genes of the isolates and the composition of the pathotypes?
We used traditional virulence analysis on wheat differential seedlings - standard GRRC (Global Rust Research Centre) procedures.
Were only ad planta tests used or also molecular biological methods?
Planta tests were used as described in section 4.2. Virulence analysis. Our study did not include molecular marker analyses.
For this question, the author is asked to explain and explain the missing information.
Relevant information was added. Please see the attachment.

Reviewer 2 Report
The manuscript ANALYSIS OF HOST-SPECIFIC DIFFERENTIATION OF PUCCINIA STRIIFORMIS IN THE SOUTH AND NORTH-WEST OF THE EUROPEAN PART OF RUSSIA presents results on the virulence diversity of yellow stripe rust isolates on different host plants and in different regions of Russia. The physiological races and virulence diversity of cereal rusts are of international importance and therefore worth to be published.
The Introduction is too short. Some information about differential sets and the history of Pst virulence development and distribution should be provided to better clarify the background and to make clearer the discussion.
The analyses are phenotype-based. As results the authors are presenting tables showing the virulence frequency and the virulence pathotypes of the Pst isolates. In addition, they show complex UPGMA dendrograms of Pst pathotypes. In my eyes, a heat map based presentation of the phenotypic data could show the results more clearly than the hard to read dendrogram presentation.
In M&M Figure 5 is superfluous. Better: an image showing the five different phenotypes (0-4).
Please find some more minor comments within the manuscript file.

Author Response
Dear reviewer,
We accepted all editorial-type changes and added some information according to your comments.
Thank you very much.
With kind regards,
Elena Gultyaeva
Answers to comments.
The Introduction is too short. Some information about differential sets and the history of Pst virulence development and distribution should be provided to better clarify the background and to make clearer the discussion.
We already responded on similar criticism of the first reviewer. Introduction was considerably extended and relevant information was added (lines 44-82, 105-124).
The analyses are phenotype-based. As results the authors are presenting tables showing the virulence frequency and the virulence pathotypes of the Pst isolates. In addition, they show complex UPGMA dendrograms of Pst pathotypes. In my eyes, a heatmap based presentation of the phenotypic data could show the results more clearly than the hard to read dendrogram presentation.
We don’t agree. Yes, sometimes a structured heatmap could be a preferable presentation of relationships between individuals. However, it is not the case here. The UPGMA dendrogram clearly reflects what we intended to show, though it is really rather hard to read it. Yet, the corresponding heatmap will be hard to read to the same extent, but its size (square matrix with names of lines and columns) is expected to be much larger. So, no changes were made.
Please, transfer information of this part (lines 195 to 210) to M&M (may be as a table…)
We don’t agree. This part better fits the discussion to provide relevant explanations.
In M&M Figure 5 is superfluous. Better: an image showing the five different phenotypes (0-4).
We removed this figure and replaced it with the requested image.
Please find some more minor comments within the manuscript file.
We accepted all editorial-type changes. Please see the attachment.

Reviewer 3 Report
The authors described the determination of the host specificity of Puccinia striiformis in different locations. This work provides new information on the host specificity of the yellow rust pathogen of wheat. The data are well presented in the manuscript. However, the following information is missing:
- Number of plants for each sample collected from the field
- How do the authors grow plants for inoculation? The details like the age of the plants, number of replicates etc are needed to include in the methods section.
- Pictures of virulence assay: the pictures need to add either in the results or in the supplementary.
Author Response
Dear reviewer,
We added some information according to your comments.
Thank you very much.
With kind regards,
Elena Gultyaeva
Answers to comments.
- Number of plants for each sample collected from the field.
In general, three or more leaves with uredia from a single cultivar or from each plot/field were considered as one sample, We added this information in lines 405-406.
- How do the authors grow plants for inoculation? The details like the age of the plants, number of replicates etc are needed to include in the methods section.
We added relevant explanations in lines 416-419.
- Pictures of virulence assay: the pictures need to add either in the results or in the supplementary.
We already responded on similar criticism of the second reviewer. The requested figure was added in the supplement).
Please see the attachment.
